# Anisotropy of Mechanical Properties and Residual Stress in Additively Manufactured 316L Specimens

**DOI:** 10.3390/ma14237176

**Published:** 2021-11-25

**Authors:** Alexey Fedorenko, Boris Fedulov, Yulia Kuzminova, Stanislav Evlashin, Oleg Staroverov, Mikhail Tretyakov, Evgeny Lomakin, Iskander Akhatov

**Affiliations:** 1Center for Design, Manufacturing and Materials, Skolkovo Institute of Science and Technology, Bolshoy Boulevard 30, bld. 1, 121205 Moscow, Russia; Yulia.Kuzminova@skoltech.ru (Y.K.); s.evlashin@skoltech.ru (S.E.); I.Akhatov@Skoltech.ru (I.A.); 2Department of Mechanics and Mathematics, Lomonosov Moscow State University, GSP-1, Leninskiye Gory 1, 119991 Moscow, Russia; fedulov.b@mail.ru (B.F.); evlomakin@yandex.ru (E.L.); 3Center of Experimental Mechanics, Perm National Research Polytechnic University, Komsomolsky Prospect 29, 614990 Perm, Russia; cem_staroverov@mail.ru (O.S.); cem_tretyakov@mail.ru (M.T.)

**Keywords:** residual stress, additive manufacturing, finite element analysis, 316L mechanical properties, mechanical testing, laser powder bed fusion (LPBF)

## Abstract

In the presented study, LPBF 316L stainless steel tensile specimens were manufactured in three different orientations for the analysis of anisotropy. The first set of specimens was built vertically on the build platform, and two other sets were oriented horizontally perpendicular to each other. Tensile test results show that mean Young’s modulus of vertically built specimens is significantly less then horizontal ones (158.7 GPa versus 198 GPa), as well as yield strength and elongation. A role of residual stress in a deviation of tensile loading diagrams is investigated as a possible explanation. Simulation of the build process on the basis of ABAQUS FEA software was used to predict residual stress in 316L cylindrical specimens. Virtual tensile test results show that residual stress affects the initial stage of the loading curve with a tendency to reduce apparent Young’s modulus, measured according to standard mechanical test methods.

## 1. Introduction

The application of laser powder bed fusion (LPBF) in industry is an opportunity to produce lightweight customized structures of complex shape with exceptional mechanical performance. However, LPBF metals are very sensitive to a number of process parameters and conditions, resulting in a great variation of possible mechanical properties depending on process setup. The understanding of influence of numerous build parameters is key to achieving the best properties, e.g., steels with high strength and ductility [1,2,3]. To characterize the anisotropy of mechanical properties of produced material, many studies use the analysis of specimens, built in different orientations [4]. In this case, an important option is to study specimens machined from produced solid parts, or to test specimens in “as-build” conditions without postprocessing.

Concerning the elastic properties, a strong correlation with the porosity level, as well as with pore size distributions and morphology is established [5]. Garlea et al. [6] used resonant ultrasound spectroscopy for elastic moduli measurement of LPBF 316L steel with a variable porosity level. They found that 9% porosity corresponds to the Young’s modulus of less than 140 GPa, while the standard value of about 200 GPa was achieved for the porosity of 0.1%. Jeon et al. [7] explained anisotropy of 316L specimens through the finite element simulation of solid media with real SEM-based porous microstructure under different loading conditions, capturing different responses depending on pore orientation and closing in compression. Mahesh et al. [8] developed a representative volume element (RVE) based on the statistical data from microstructure characterization, including grain size, the crystallographic orientations, and porosity. They applied a crystal plasticity model with damage evolution for the justification of the hypothesis of influence of the pore shape on anisotropy.

However, some studies report a deviation of LPBF elastic moduli from the conventional value of very low porosity [9], so a different interpretation is required. Niendorf et al. [10] associated almost a double decrease of the Young’s modulus with {001} texture alongside the build direction during processing. The effect of modulus decrease in build direction was also observed in [11] for the IN738LC superalloy, and authors extracted crystallographic texture from electron backscatter diffraction in order to estimate Young’s modulus using Voigt, Reuss and Hill’s methods [12]. Another microstructural explanation of anisotropy accompanied by higher mechanical properties in the horizontal direction in terms of grain aspect ratio and orientation is presented in [13].

Charmi et al. [14] presented a detailed study of 316L anisotropy, reporting the highest Young’s modulus of 225 GPa for horizontal specimens, and the lowest of 180 GPa for vertical ones, measured by the resonance method. In the mentioned work, a tensile test showed the maximum modulus of 215 ± 3 GPa in the vertical direction, and the minimum was 192 ± 7 GPa in the vertical, but the authors were concerned about possible inaccuracies of the extensometer configured for the large strains. The authors concluded that elastic anisotropy is driven by a crystallographic texture, since it was captured using single-crystal elastic constants as inputs for the MTEX [12] simulations. They indicate residual stresses as negligible factors for elastic and plastic anisotropy. However, the specimens were machined from the prismatic solids with consequent soft heat treatment, and presented residual stresses are significantly lower than in the case of “as-build” properties.

Röttger et al. [15] studied vertically and horizontally oriented 316L specimens built with different process parameters, and while the reduction of Young’s moduli in comparison with conventional steel was observed in many cases, they indicate neither texture nor porosity as the only dominant factor. The authors also assume that reversible dislocation movements cause a reversible elastic expansion contribution to the total elongation, so the apparent Young’s modulus decreases. The authors also carried out a hot isostatic pressing of the horizontal specimen at 1150 °C, which led to a significant increase of Young’s modulus from 165 GPa to 205 GPa. Eventually, they discussed mechanical properties with respect to a reduced dislocation density, a homogeneous distribution of the elements, and a crack-free microstructure after treatment. In contrast to some studies, the authors obtained the highest modulus of as-build specimens in a vertical direction. The results with high moduli of vertical specimens in as-build conditions can also be found elsewhere [16], with some remarks about problematic measurement accuracy.

Zhang et al. [17] compared tensile diagrams of vertical and horizontal 316L dog-bone specimens in as-build conditions, establishing higher Young’s modulus and yield strength of horizontal ones. Nevertheless, their build method with preheating at 150 °C for vertical specimens allowed them to surpass properties of horizontally oriented ones. The specimen produced without preheating was significantly more distorted during processing, and the authors identify that the effect of residual stress is evident for this case.

Heat treatment allows for the relieving of residual stresses, but a degree of influence on microstructure may vary depending on temperature and holding time. In other words, it is problematic to use a heat treatment with the aim of obtaining residual stress-relieved specimens without an effect on the microstructure. The most common methods for residual stress measurements are neutron [18,19] and X-ray diffraction [20]. Both methods require identification of the microstructure in a stress-free state [21], and X-ray diffraction is additionally strictly limited by penetration depth and sensitivity to surface treatment.

Chao et al. [22] investigated the evolution of microstructure, residual stress and resultant mechanical performance of SLM (Selective Laser Melting) processed 316L after a heat treatment over a wide temperature range of 400–1400 °C, reporting gradual microstructure change with temperature increase. Suryawanshi [23] found anisotropy in tensile tests of specimens after 1 h of heat treatment at 700 °C, identifying that this level of treatment does not correspond to a complete homogenization. Ronneberg et al. [24] used heat treatment of LPBF 316L steel to isolate the influence of porosity and microstructure on the anisotropy of mechanical characteristics, categorizing heat treatment in three ranges: recovery, homogenization, and annealing. The annealing fully recrystallizes the microstructure with isotropic properties. The authors conclude that porosity does not cause anisotropic yield behavior, which contradicts the aforementioned references [7,8].

While many studies do not consider the influence of residual stresses on mechanical properties, Chen et al. [25] used the crystal plasticity model for the explanation of tension-compression asymmetry through the simulation of residual microstresses [26] in SLM 316L. Furthermore, experimental studies [27,28] show residual stresses close to yield strength, which cannot be ignored in the case of mechanical loading.

Although many works found mechanical anisotropy in LPBF processed materials, it was problematic for us to find a comprehensive study whether residual stress in test specimens is capable of misrepresenting the characteristics of the material with a standard test procedure. Since the presence of defects, pores and microstructural specificity mainly affect yield and strength characteristics of the material, such an essential deviation of elastic properties can be the key for understanding of main factors influencing anisotropy. This motivates us to investigate the influence of residual stresses on anisotropy focusing on elastic response, as the residual stresses are suggested to affect mainly the initial stage of loading. A coupled thermo-mechanical finite element analysis of standard tensile specimen during layer-by-layer processing is used for the computation of residual stresses [29,30,31,32,33,34,35]. This approach is associated with macro stresses, and ignores the physics of the melting pool and consolidation, but is computationally effective to capture the residual stress distribution in the scale of the specimen. The ‘block dump’ activation of elements is used, where every block corresponds to tens of physical layers [36], and every newly activated block evolves a temperature history, determined in the heat transfer analysis step. For the validation of the modeling technique, an experiment with a partial cut of the as-built cylindrical rod along the midplane was carried out, so halves of the rod deviated due to residual stresses. Then, the proposed simulation of LPBF processing was performed for the both vertical and horizontal orientations of tensile specimens for the evaluation of residual stresses and their degree of influence on the mechanical behavior. In particular, a tensile virtual test of specimens with the presence of residual stresses was conducted and compared with a stress—free assumption focusing on Young’s modulus determination.

## 2. Materials and Experimental Methods

### 2.1. Selection and Preparation of the Specimens

An experimental set is schematically represented in Figure 1a and consists of a subset of vertically oriented (“V”) specimens and two subsets of horizontally oriented tensile specimens, indicated as “H_0” and “H_90”, depending on orientation in the plane of build platform. In addition, one vertically oriented cylindrical specimen is used as a validation experiment for the LPBF simulation, which is simpler than analysis of complex cantilevers [37,38,39] and allows for the estimation of residual stresses of the tensile specimen with a similar cylindrical gage zone. The X-Y square scan strategy with the base pattern size of 4.0 mm is applied with the rotation of every subsequent layer by 90° with respect to the previous one (Figure 1b), as described in [40]. Pattern formation is implemented in a checkerboard order, as shown in the example of Figure 1c for the case of the H_0 specimen, and the base square pattern is truncated following target geometry boundaries.

The geometry of cylindrical tensile specimens is selected according to ISO 6892-1 (Figure 2). While vertical specimens are appropriate for testing with the gage zone in as-built conditions, i.e., without additional machining, horizontal specimens have to be relieved from large building supports alongside the full length. Thus, all vertical specimens were considered in as-built conditions and were not subjected to mechanical or thermal processing, excluding separation from the build platform and threading for tensile fixture. In contrast, horizontal specimens were subjected to sufficient machining, which is further considered in the context of residual stresses.

The powder of 316L stainless steel (Oerlikon, Freienbach, Switzerland) was used for the building of all specimens. A manufacturing process was conducted using the 3D metal printer TruPrint 1000 (Trumpf) with parameters according to manufacturer recommendations for 316L steel, summarized in Table 1. The relative porosity of produced parts analyzed with the optical microscope Axio Scope.A1 (Carl Zeiss, Jena, Germany), according to ASTM E1245, does not exceed 0.1%.

### 2.2. Tensile Tests

Tensile tests were carried out in accordance with ISO 6892-1 standard on an Instron 5969 machine. The loading rate was 0.001 s^−1^. Strain measurements were done using the virtual extensometer method from a digital image correlation system (Correlated Solutions, Irmo, SC, USA) based on a gauge length of 36 mm. Full experimental tensile diagrams for all three subsets are shown in Figure 3, and an initial part of the diagrams below 1% strain is presented in detail in Figure 4. Notice that diagrams in Figure 4 do not depict raw data directly from the testing system, but rather the refined one from the noise using the filtration procedure and shifted to zero stress point due to a slight initial offset. The mechanical characteristics are summarized in Table 2 for vertical (“V”) set, in Table 3 for “H_0” set, and in Table 4 for “H_90” set.

From tensile test results in Figure 3 and Figure 4, the difference of mechanical response between horizontal and vertical sets is clearly observed, while there is no significant difference between H_0 and H_90 sets of specimens. Moreover, a variation of characteristics between the specimens of the vertical set is higher. The average Young’s modulus of vertical specimens is 158.7 GPa, which is significantly lower than the moduli of horizontal sets of 196 GPa and 199 GPa, whose values are typical for conventionally produced steel. Yield tensile strength (YTS) and ultimate tensile strength (UTS) characteristics of horizontal sets also exceed ones for the vertical set. The ductility of vertically built specimens is slightly higher and is within 5%.

### 2.3. Validation Experiment

The cylindrical rod with a diameter of 8.0 mm and length of 80.0 mm was built vertically. After removal from the chamber, the rod was sectioned from the build plate, and then a cut by 0.2 mm wire EDM along the midplane was carried out starting from the bottom base (Figure 5). An Accutom-100 machine with a cooling water system was used for the cut. The tip of the cut is located 5.85 mm away from the top base of the cylinder. After the cut, two pairs of corner vertices on the bottom surface deviated to the distances of 7.84 mm and 8.17 mm from each other, as shown in Figure 5.

## 3. Simulation Approach

### 3.1. General Details

Simulation of the LPBF process is implemented via a special module of the Abaqus FEM software for additive manufacturing. The first step involves transient heat transfer FEA during layer-by-layer activation of elements, which corresponds to deposition of the powder layer, with a subsequent SLM process. The scheme of the approach is shown in Figure 6. Recent studies show an opportunity to replace a detailed modelling of laser scanning on a microscale level by a point-based heat source concept and using a block-dumping approach to reduce unfeasible computational time [36]. A mesh sensitivity study of particular cases shows [41], that a Gauss-type [42] or Goldak laser beam model can be reasonably replaced by an equivalent point-based heating source with a great reduction of the number of elements in the mesh, so a detailed simulation of local instantaneous temperature peaks under laser scanning is not critically important for the accuracy of mechanical analysis step.

The obtained thermal history during the build process is used for the static mechanical analysis step with the similar activation of elements. The element is initially activated at a certain ‘stress-free’ temperature T_sf_, and is cooled down to a temperature obtained on a corresponding increment of thermal analysis step. Thus, the temperature of the element in the static analysis step after activation evolves according to the temperature history from the thermal step, and element shrinkage causes internal stresses. The ‘stress-free’ temperature T_sf_ has no clear physical definition, and commonly is defined as the melting temperature, but it can be numerically or experimentally calibrated [41]. The lack of data for modelling, e.g., mechanical characteristics at an elevated temperature, can be resolved by the use of a combination of conventional properties taken from the literature and measured on-site for produced material.

Considering the mechanical analysis step, the process is driven by thermal shrinkage with the temperature change ∆T, inducing thermal strain $\varepsilon_{thermal}$. It requires the input of temperature dependencies of elastic moduli and thermal expansion coefficient α of isotropic material, which in this work are taken from [43] for conventional steel. Eventually, the basic relations are formulated as follows:(1)$$\varepsilon_{thermal}=\alpha ∆T,$$
(2)$$\varepsilon_{total}=\varepsilon_{elastic}+\varepsilon_{plastic}+\varepsilon_{thermal},$$
(3)$$\sigma =C\left(T\right):\varepsilon_{elastic}\;$$
(4)$$f_{yield}=\sqrt{\frac{2}{3}S:S}-\sigma_{0}\left(\varepsilon_{eq},T\right)\;,$$
where $f_{yield}=0$ is yield condition, elastic stiffness tensor $C\left(T\right)$ is temperature-dependent, $\varepsilon_{eq}$ is equivalent plastic strain, S is the stress deviator used in von Mises yield criterion with the yield stress $\sigma_{0}$, defined as a function of $\varepsilon_{eq}$ and temperature T. In this work dependencies for $\sigma_{0}\left(\varepsilon_{eq},T\right)$ in Equation (4) for T = 20 °C are based on experimental diagrams for horizontal specimens (Figure 3) with transformation into true stress-strain dependency. The choice of diagrams only for horizontal specimens is questionable, but is caused by our hypothesis of the significant residual stresses in vertical specimens, which misrepresent experimental results. In other words, mechanical characteristics from residual stress-relieved specimens are more appropriate. This statement will be discussed further in this work with the support of a validation experiment. Similarly, Young’s modulus of 198 GPa is used for the elastic tensor formulation.

There is a lack of experimental data on hardening at an elevated temperature for the 316L steel. Therefore, temperature scaling is assumed to be the same as for conventionally manufactured 316L material [43] and is defined with respect to melting temperature T_m_ = 1400 °C and reference temperature T_0_ = 20 °C. The diagrams used in the simulation are presented in Figure 7.

### 3.2. Finite Element Model

Following the mentioned block-dumping approach, the element height for meshing of the cylindrical rod is chosen as 0.4 mm, which corresponds to ~20 physical layers in one element. The number of hexagonal elements in a cross section is 390, so the total number of elements in the model for the entire cylinder with a height of 80 mm is 78,000 (Figure 8a,b). A preliminary heat transfer modelling showed that after deposition of each layer the conduction of heat occurs rapidly. This effect is caused by a simple geometry and a sufficient time period between layer deposition for the heat outflow. Thus, the thermal gradient within the sample and substrate is relatively small in comparison with a great temperature drop from the ‘stress-free’ state instantly after laser heating. For this reason, it is effective to skip a direct thermal analysis and to use the ‘eigenstrain’ method. This method involves only the static mechanical analysis for sequential shrinkage of every newly activated layer from T_sf_ = 850 °C to the temperature of the substrate at the end of the pattern scanning, equal to 80 °C (Figure 8b). The proposed assumptions allow us to ignore the build platform in the analysis, so the bottom surface of the cylinder is assumed to be fixed in all degrees of freedom. In addition, direct thermal analysis is faced with uncertainties in the heat transfer parameters, which are difficult to measure in the manufacturing chamber, e.g., convection and emissivity.

The characteristic element size in FEM of the vertical tensile specimen is 0.4 mm, the length is 40 mm and the diameter is 6 mm to match gage dimensions of the tensile specimen (Figure 9a). In contrast, the horizontally built tensile specimen suggests a considerable machining after LPBF processing due to the adjustment to the build platform alongside the specimen (Figure 9b). In this case, the only analysis is residual stress prediction in the specimen of such form directly after sectioning from the built platform, since the simulation of the machining is not rational. The scheme of element activation is shown in Figure 1c, following the base square pattern of 4 mm.

## 4. Simulation Results

### 4.1. Simulation Results of Validation Problem

Distribution of residual stresses in the cross section of the build cylindrical rod after separation from the build platform and before the longitudinal cut is shown in Figure 10. The complex stress state is clearly observed: the radial and circumferential stress components reach relatively high values, but they are lower than the tensile yield limit. However, the axial stress component along the build Z-direction is significantly above the plasticity limit, and the region in the vicinity of the outer surface is in tension, while the interior is in compression. It is interesting to note that this distribution is opposite to the quenching process, when, due to the nonuniform cooling, areas near the outer surface become compressed, and the interior is under tension [44]. Peak values of shear components are about one order of magnitude less and are not presented. The equivalent von Mises stress distribution in Figure 10d shows that plasticity prevails within the entire part with a sharp transition zone between the outer and interior zones. These results are qualitatively similar to the residual stress distribution published in [45,46], including experimental measurements.

As is shown in Figure 11, the stress distribution within the as-build part is similar in every cross section, which is located sufficiently far from the end faces, and affected by boundary conditions. Thus, a transversal cross section in the middle of the rod is selected to study a stress state along the radial path, as shown in Figure 10. To characterize the stress state, a triaxiality parameter $\xi =-p/\sigma_{0}$ is introduced, where $p=-\sigma_{ii}/3$ is the hydrostatic stress, and $\sigma_{0}$ is the von Mises stress. It can be concluded from the plots of Figure 11 that the triaxiality parameter is strongly driven by uniaxial stress $\sigma_{Z}$. Moreover, the interval 2–6 mm from the path is in a predominant biaxial compression state (ξ≈−0.6), and outer regions are under the condition of biaxial tension (ξ≈+0.6).

After the build process has been modelled, a simulation of longitudinal cut was performed by the removal of a row of elements along the midplane. In addition, the fixed boundary condition on the bottom face was released in FEM, which corresponds to sectioning from substrate. This boundary condition release was compensated by the introduction of an artificial inertia force to satisfy equilibrium. The thickness of the removed row is 0.4 mm and is in approximate accordance with the thickness of the eroded material in the experiment. Element removal was performed in static assumption, and there are no significant differences between the instant cut of the entire row or in element-by-element removal. After the cut, the axial stresses were slightly decreased, as shown in Figure 12. The distance between the deviated vertices on cut surface is 7.9 mm (Figure 13), and almost exactly coincides with the experiment (Figure 5).

### 4.2. Residual Stress in Horizontal Specimen

The von Mises equivalent stress distribution within the longitudinal section of the horizontally built specimen (Figure 9b) after release from the build platform is shown in Figure 14. Compared with the results of Figure 10 for the vertical specimen, one can conclude that the overall stress level is drastically lower in this case. The major region of the built part is in the range of equivalent stress at 200–300 MPa, and only the thin region near the outer surface is in a stress state close to the yield limit. In addition, the real specimen was machined in order to obtain circular shape, so the final expected residual stresses decreased. The visible periodicity in distribution is caused by the checkerboard scanning strategy.

### 4.3. Virtual Tensile Test of Vertical Specimen

To study the effect of residual stresses in the vertical specimen on the loading diagram, the virtual tensile test was performed using the proposed model. After simulation of layer-by-layer processing and separation from the build platform, boundary conditions were re-applied in restart analysis to simulate tension. The simulation results of tension without consideration of residual stresses are also presented. A comparison of tensile loading curves within 1% elongation were obtained through simulation and experiment, and are shown in Figure 15. Notice that the presented curve for residual stress-free conditions (the green line of Figure 15) is also associated with the test for horizontal specimens, since it is used as an input for simulation. The presence of the residual stress after the build process causes the reduction of the slope angle of the loading diagram at the initial stage of loading in comparison to the analysis of the residual stress-free specimen. In other words, the residual stress affects the measurements of Young’s modulus, and the presented virtual test gives the approximate value of 150 GPa, while the experimental mean value is 158.7 GPa for the vertical specimen. The simulation under residual stress-free assumption gives the one used in the model, i.e., 198 GPa, which is a common result for conventionally produced steel.

The evolution of total axial stress in the cross section of the test sample is illustrated for three selected levels of load. The presented stress value corresponds to a sum of residual stress in the axial direction, and for the axial stress due to external tensile force. The compression of the interior of the specimen is gradually compensated by the increase of external tension and a reduction of tensile stress near the outer surface, so at 1% tensile strain only a weak stress gradient remains (point 3 of Figure 15).

## 5. Conclusions

In this study an established modelling approach was implemented to estimate residual stresses in cylindrical 316L specimens, built in horizontal and vertical orientations. The validation experiment with the distortion of a specimen with cut was performed and compared to simulation. The simulation revealed significant residual stress components with values near the yield limit for the vertically built specimens. The overall residual stress level in horizontal specimens is at least three times lower than for the vertical one. A virtual tensile test of the cylindrical sample was performed with the assumption of residual stresses present after the build process. The apparent Young’s modulus, measured in the specimen with the presence of residual stresses, is reduced to 25% in comparison with the value of 198 GPa at room temperature, obtained with a residual stress-free assumption. The physical tests of 316 steel specimens show a similar reduction of Young’s modulus for vertically oriented specimens, and obtained plasticity characteristics are also reduced. The horizontal specimens were sufficiently affected by machining due to the removal of the build substrate, and this is considered as an additional stress relieving factor. Thus, the residual stress is a potential explanation of the apparent anisotropy of elastic properties, and it should be considered in conjunction with other factors of anisotropy, such as porosity and microstructure. Moreover, this fact complicates the use of “in-house” mechanical characteristics for the simulation, since in the presented case, test results can be incorrectly interpreted as the true anisotropy of mechanical properties.

## Figures and Tables

**Figure 1 materials-14-07176-f001:**
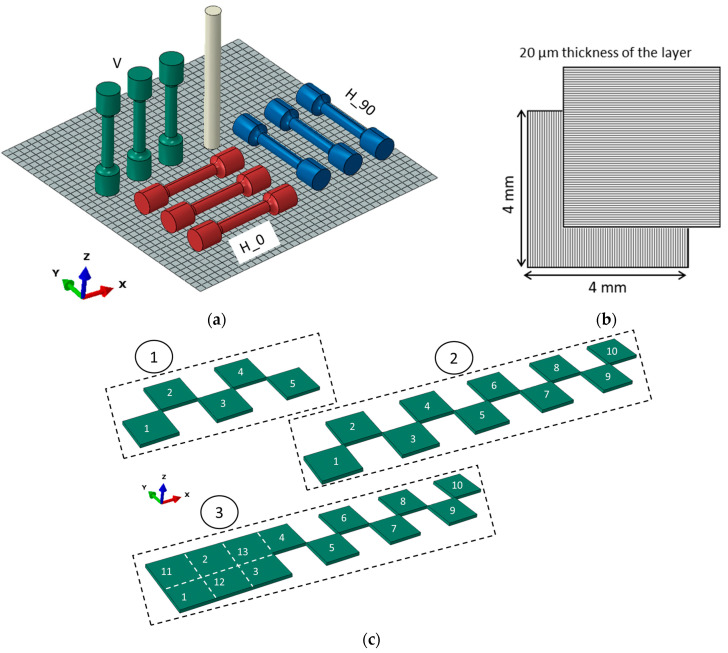
(**a**) Scheme of experimental set on build platform, (**b**) square pattern, (**c**) example of patterns formation sequence for the horizontal H_0 specimens.

**Figure 2 materials-14-07176-f002:**
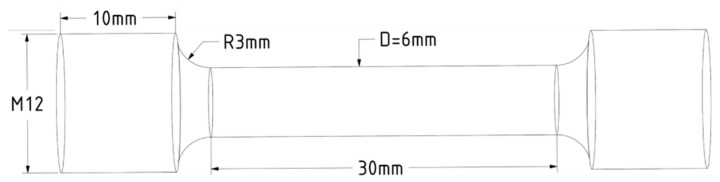
Dimensions of a cylindrical tensile specimen.

**Figure 3 materials-14-07176-f003:**
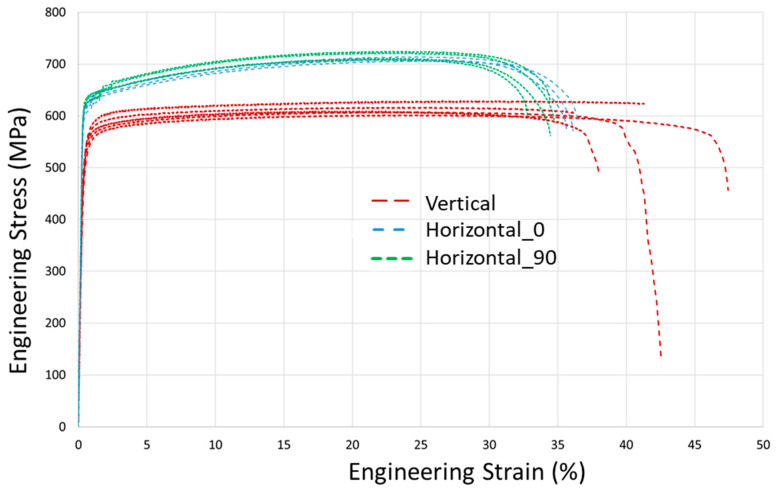
Tensile test results of LPBF processed as-build 316L samples.

**Figure 4 materials-14-07176-f004:**
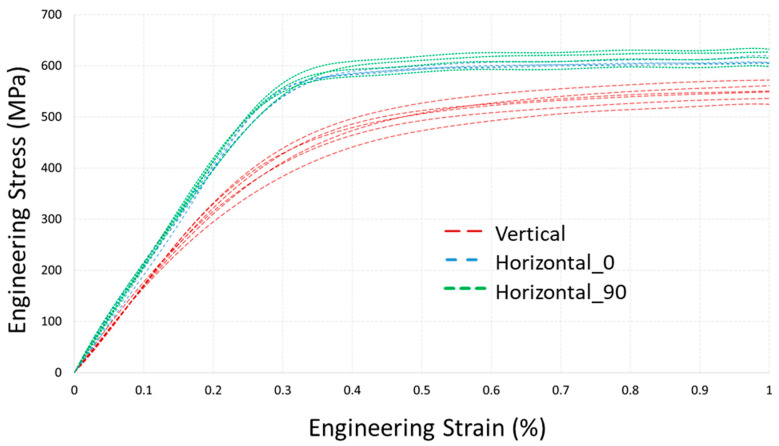
Tensile test results of LPBF processed as-build 316L samples.

**Figure 5 materials-14-07176-f005:**
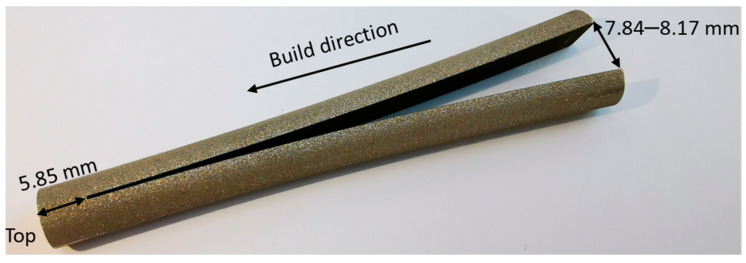
Validation experiment with cut of vertically built cylindrical rod.

**Figure 6 materials-14-07176-f006:**
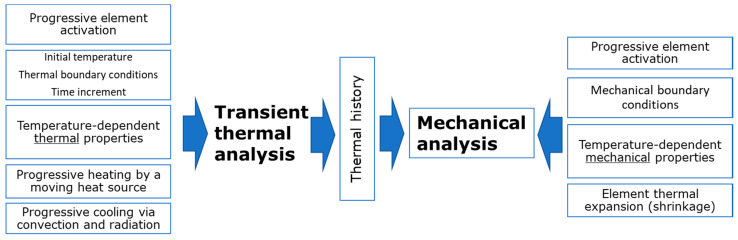
Scheme of thermomechanical simulation of SLM process.

**Figure 7 materials-14-07176-f007:**
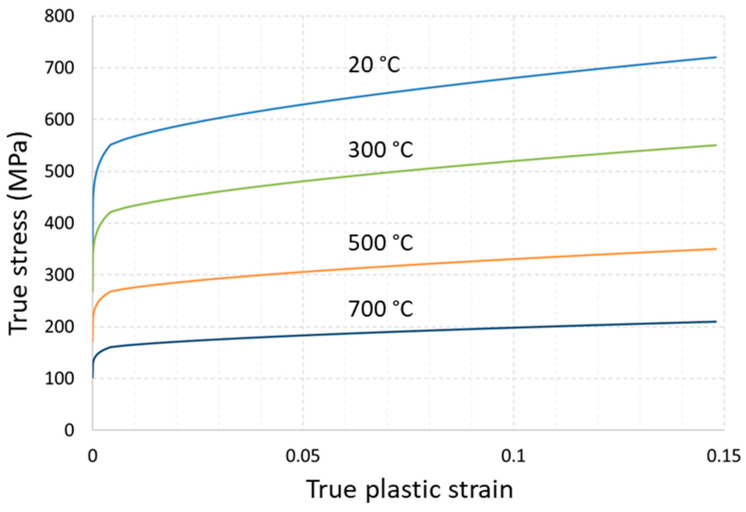
Temperature scaling of modified hardening behavior (MPa). Data for 20 °C are based on tensile experiments for horizontal sets.

**Figure 8 materials-14-07176-f008:**
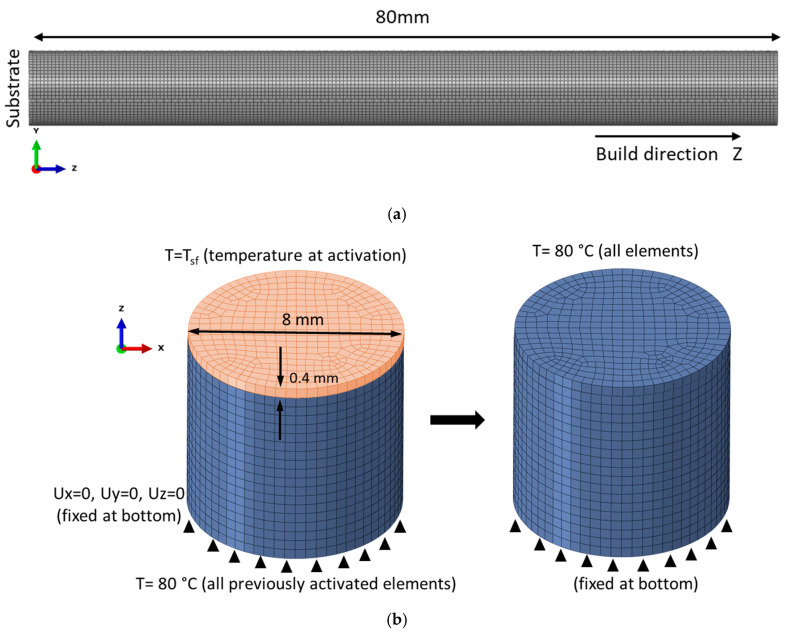
(**a**) Mesh of the cylindrical specimen built in vertical direction, (**b**) scheme of elements activation and boundary conditions.

**Figure 9 materials-14-07176-f009:**
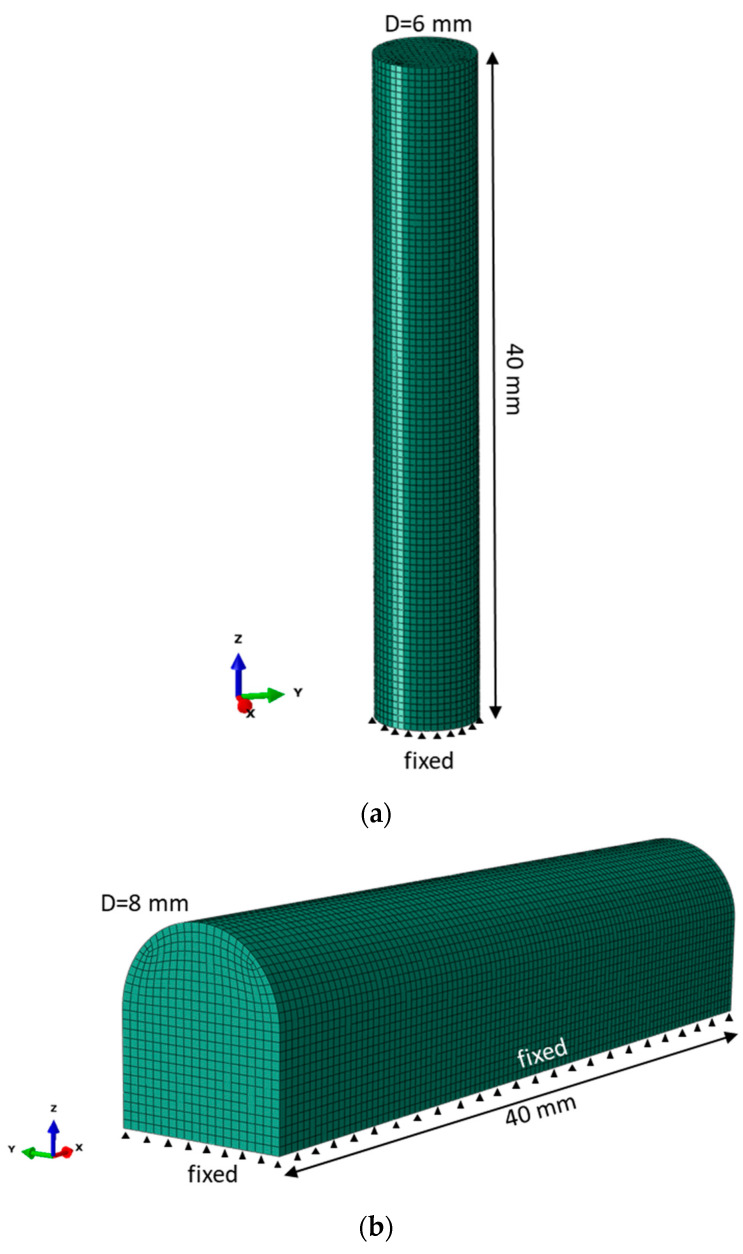
(**a**) Dimensions, boundary conditions and mesh of vertically oriented specimen, (**b**) dimensions, boundary conditions and mesh of horizontally oriented specimen.

**Figure 10 materials-14-07176-f010:**
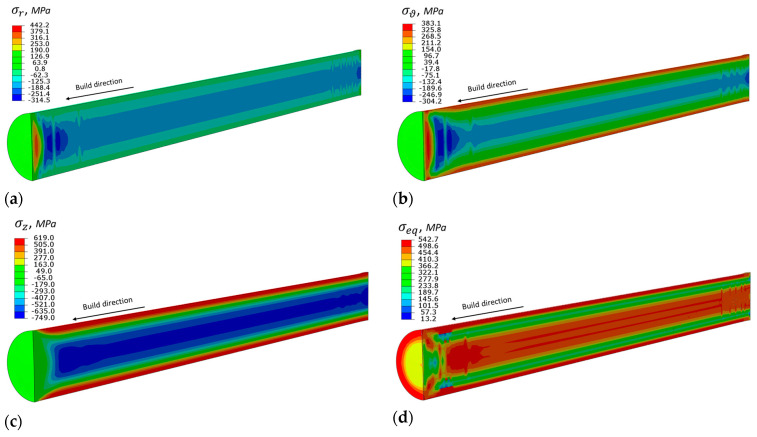
Residual stress distribution in as-build cylindrical rod before cut (MPa). (**a**) radial component, (**b**) circumferential component, (**c**) axial component along build direction, (**d**) von Mises equivalent stress.

**Figure 11 materials-14-07176-f011:**
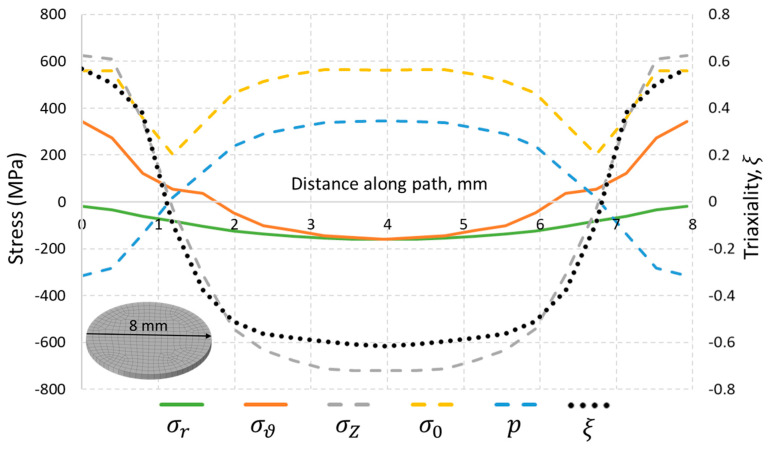
Stress characteristics along the diametral path.

**Figure 12 materials-14-07176-f012:**
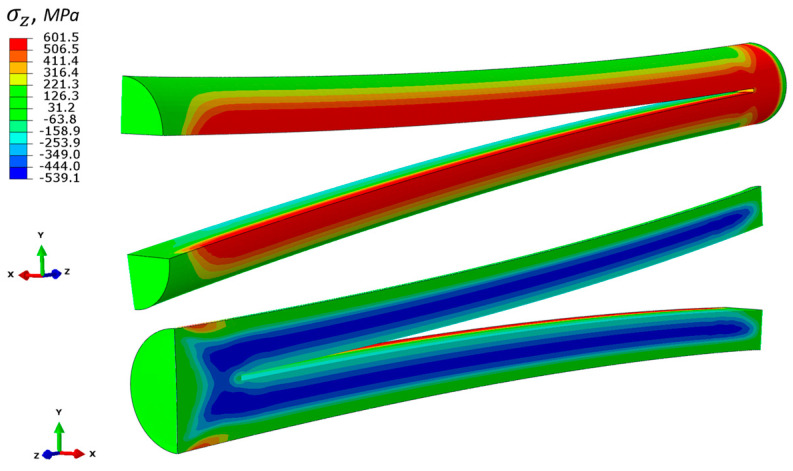
Axial residual stress distribution within the cylindrical rod after cut (MPa).

**Figure 13 materials-14-07176-f013:**
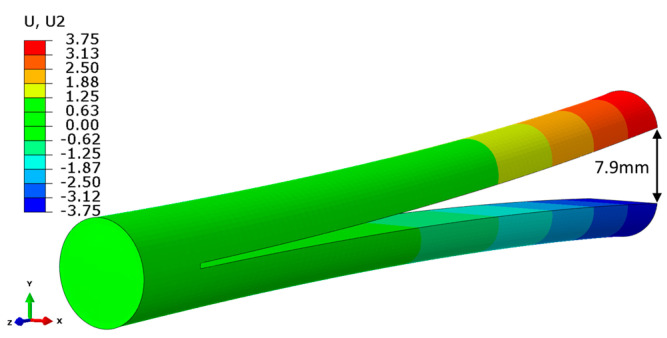
Deflection from the midplane (mm). Total distance between vertices is 7.9 mm with the account of removed material thickness of 0.4 mm.

**Figure 14 materials-14-07176-f014:**
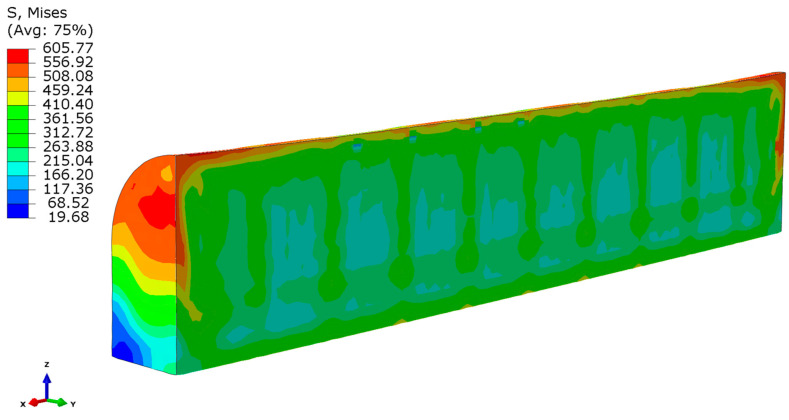
Equivalent stress distribution in horizontally built specimen (MPa).

**Figure 15 materials-14-07176-f015:**
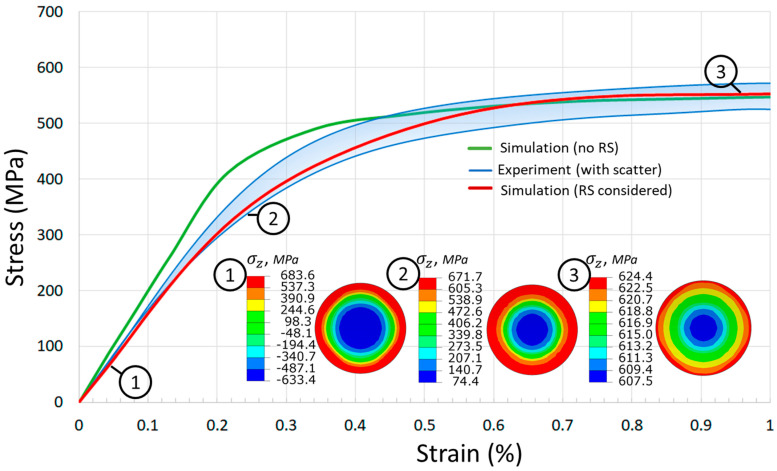
Effect of residual stress presence in the simulation of tensile test.

**Table 1 materials-14-07176-t001:** Build parameters.

PBF Parameter	Values
Laser power	113 W
Laser spot diameter	55 µm
Hatch spacing	80 µm
Layer thickness	20 µm
Laser scan speed	750 mm/s
Gas speed (Ar)	2.5 m/s
Oxygen level	<0.3 at. %
Pressure in chamber	1 bar

**Table 2 materials-14-07176-t002:** The measured mechanical properties of “V” set.

Spec. #	Young’s Modulus, GPa	YTS, MPa (0.2% Offset)	UTS, MPa	Elongation at Fracture, %	Reduction of Cross-Sectional Area at Fracture, %
V_1	160	535	630	41	48
V_2	163	535	615	37	46
V_3	157	530	610	38	43
V_4	161	535	615	46	42
V_5	159	525	605	36	40
V_6	152	510	600	36	41
Average	158.7	528.3	612.5	39.0	43.3
Standard deviation	3.8	9.8	10.4	3.6	3.1

**Table 3 materials-14-07176-t003:** The measured mechanical properties of “H_0” set.

Spec. #	Young’s Modulus, GPa	YTS, MPa (0.2% Offset)	UTS, MPa	Elongation at Fracture, %	Reduction of Cross-Sectional Area at Fracture, %
H_0_1	200	630	725	32	58
H_0_2	198	615	710	31	55
H_0_3	201	610	705	30	56
H_0_4	185	610	705	30	54
Average	196	616.2	711.2	30.8	55.8
Standard deviation	7.4	9.5	9.5	0.8	1.7

**Table 4 materials-14-07176-t004:** The measured mechanical properties of “H_90” set.

Spec. #	Young’s Modulus, GPa	YTS, MPa 1234567 (0.2% Offset)	UTS, MPa	Elongation at Fracture, %	Reduction of Cross-Sectional Area at Fracture, %
H_90_1	199	615	705	29	52
H_90_2	211	620	705	30	58
H_90_3	188	605	695	28	56
H_90_4	198	610	700	28	49
Average	199.0	612.5	701.3	28.8	53.7
Standard deviation	9.4	6.5	4.8	0.8	4.3

## Data Availability

Raw data are available from corresponding author upon reasonable request.

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
