# Peer review of "Anisotropy of Mechanical Properties and Residual Stress in Additively Manufactured 316L Specimens"

_materials, 2021, doi:10.3390/ma14237176_

Round 1

Reviewer 1 Report

The presented work "Anisotropy of mechanical properties and residual stress in additively manufactured 316L specimens" is devoted to an essential topic of understanding physical mechanisms that occur in additively manufactured materials. 
It is well written and organized manuscript. Especially the part of the simulation of residual stresses in printed steel. 

I see just several small issues:

1. I'm not sure that "cylindrical pillar" is an optimal term for such shaped parts. Please, consider replacing it with "a rod" or "a cylinder". 
2. Can you emphasize why it is important to understand the mechanism of residual stress formation if anyway laser-printed parts overgo heat treatment for stress relief?
3. How your simulation approach can be used for complex shape parts?

Reviewer 2 Report

In the manuscript “Anisotropy of mechanical properties and residual stress in additively manufactured 316L specimens” the authors report on the known effect of anisotropy in the direction-dependent manufacturing process and examine the residual stress as a primary source for these differences, expressed in variations in Young’s modulus. Their explanations of computational modeling are clear and sound. The work is well-written and scientifically justified since there is still no agreement on the origin of anisotropy in the community of additive manufacturing. However, I would suggest a minor revision considering the following issues:

  1. Supplying more information and highlighting the major causes that lead to anisotropy, especially those that already have been proved such as high angle grain boundaries, subgrain dendritic microstructures, crystallographic texture, etc.
  2. Revise the figures and their content. E.g., the purpose of figure 1c is not clear. Are its presence and size justified? Figures 2, 6 are low quality and should be replaced, Are figures 8 and 9 justified? (not very informative and can be moved to the supporting information section)

I recommend this paper for publication after the minor revision.

Reviewer 3 Report

Nice piece of work! Just couple of small things:

Page 7 of 19, please capitalize 'elongation' and 'reduction"

Confused with the equation of (1,c) and (1,d) - what does ":" mean in the eqn.?

Please check all the figure caption, keep the way of presenting units consistent, such as Fig 15 (MPa) and , %
